# Health Status and Nutritional Habits in Maldives Pediatric Population: A Cross-Sectional Study

**DOI:** 10.3390/ijerph192315728

**Published:** 2022-11-25

**Authors:** Emanuela Cazzaniga, Antonina Orlando, Annalisa Terenzio, Carlotta Suardi, Chiara Mognetti, Francesca Gennaro, Laura Antolini, Paola Palestini

**Affiliations:** 1School of Medicine and Surgery, University of Milano-Bicocca, 20126 Monza, Italy; 2Bicocca Center of Science and Technology for Food, University of Milano-Bicocca, Piazza della Scienza, 2, 20126 Milano, Italy; 3Cardiologic Unit, Istituto Auxologico Italiano, IRCCS, 20100 Milan, Italy; 4Department of Experimental Oncology, Istituto Europeo di Oncologia (IEO), European Institute of Oncology, Istituto di Ricovero e Cura a Carattere Scientifico (IRCCS), 20133 Milano, Italy

**Keywords:** excess weight, food habits, hypertension, children, health status

## Abstract

Chronic noncommunicable diseases (NCDs) have become the major cause of morbidity and mortality in the Maldives, triggered by the nutrition transition to a “Western diet” that dramatically increases the prevalence of excess weight and hypertension. Our study aimed to evaluate dietary habits, blood pressure (BP) and body mass index in Maghoodoo Public School’s students. A sample of 145 students (72 males and 73 females, age 9.37 ± 2.97 years) was enrolled. Factors causing excess weight were investigated through descriptive statistics. The relationship between blood pressure percentiles and possible influencing factors was investigated by a linear regression model.. Excess weight was present in 15.07% and 12.5% females and males, respectively. 15.18% of the subjects had elevated BP, with a significant difference according to gender detected only in the PAS *z*-score. Eating habits were investigated through a parent-filled questionnaire; 70.15% of the students consumed less than two portions of fruit per day, with a significant difference between gender (84.06% and 55.38% for boys and girls, respectively, *p* < 0.0001) and 71.64% ate less than two servings of vegetables per day. An alarming finding emerged for sweet snacks (30.6% of the students consumed 2–3 servings per day) and sugary drinks (2–3 servings per day for 32.84% of students) consumption. Our findings suggest that excess weight and hypertension in this population could be due to energy-rich, packaged-foods consumption. A nutrition education approach might thus help to reduce cardiovascular risk.

## 1. Introduction

The Maldivian Archipelago is located in the Indian Ocean, 600 km south of the Indian subcontinent. The country lacks land-based natural and mineral resources, a fact which makes the entire economic production highly dependent on imports, with a consequent heavy dependence on foreign exchange. Indeed, intensive agricultural production is limited because of poor soil quality and scarcity of freshwater. Therefore, most of the staple food items are imported [1]. In parallel to lifestyle changes associated with the increasing prevalence of risk factors, such as consumption of sugary drinks and sugary and fatty foods, together with sedentary lifestyles, chronic noncommunicable diseases (NCDs) have emerged as the major cause of morbidity and mortality in the country [2]. Eighty-one percent of deaths are attributed to NCDs, thus representing the leading cause of mortality in the Maldives; in addition, as detected by the 2014 Global Health Observatory data, this frequency is the highest in the region [3]. Owing to the burden on society, prevention of NCDs has become a priority for the Maldivian government [1].

Dietary habits, high blood pressure, and increased body mass index are in the top five risk factors for disease burden in the Maldivian country [4]. In Western countries, cardiovascular risk factors are present both in adults [5] and in children [6,7], whereas, up to now, they have been detected as a significant health problem in South Asian countries only in adults [8]. Since the prevalence of risk factors associated with NCDs in children and adolescents in these countries is mainly unknown, it is crucial to investigate the prevalence of obesity and hypertension and its determinants in pediatric age, in particular, in the Maldive Islands [9].

The nutrition transition to a “Western diet” dramatically increases overweight, obesity, hypertension, and NCDs [10,11]. The emergence of this double burden is attributed to the nutrition transition that commonly follows rapid economic development characterized by fast secular trends (e.g., migration and urbanization). These conditions lead to low levels of physical activity and high consumption of refined, energy-dense foods without the complete elimination of undernutrition [12].

The health and nutritional status of the pediatric population in the Maldives needs improvement, since children and adolescents are torn between the influences of Western culture and the practicalities of living in the Maldives, where food sources are limited and unpredictable [13,14]. It must also be pointed out that undernutrition still remains a problem in the Maldives, due to a restricted diet and lack of access to fruits and vegetables. A survey performed in 1999 reported that 51% of all the measured children were below the cutoff point of the weight/age ratio, thus indicating malnutrition [15]. On the other end, Western junk food has strongly influenced Maldivian children and adolescents [16], and young children remain particularly vulnerable to malnutrition as nutrition transitions progress. These observations raise the question about the adequacy of the diet and its ability to provide healthy growth [17].

It is widely recognized that westernized populations are plagued by many “civilization diseases” such as cardiovascular diseases, pathologies that are still rare in nonwesternized people. There is a growing awareness that the cause of this great discrepancy lays in the deep and fast changes in diet and lifestyle of these populations [18]. For these reasons, a specific objective in the Multisectoral NCDs Action Plan of Maldives is the reduction in modifiable risk factors for noncommunicable diseases, and of the underlying social determinants, through the creation of health-promoting environments [4]. In this perspective, health and dietary data would be helpful to assess in a quantifiable manner the contribution of Western foods (rich in sugar, fat, and salt) to the diet of the Maldivians.

This cross-sectional study aimed to analyze the current nutritional status of children and adolescents living in one Maldivian Island (Magoodhoo, Faafu Atoll), in order to evaluate the influence of feeding patterns, starting from breastfeeding to the current ones. We also compared the results obtained in males and females to assess possible gender differences in dietary habits; these data could be important since these habits, including pediatric ones, have been shown to differentially affect CVD risk in men and women. In addition, we also investigated the timing of salt and sugar introduction in their diets. All the collected data are helpful to better understand the possible link between the dietary change that occurred during the last few years (i.e., the preference for a Western diet) and the higher prevalence of hypertension and excess weight in pediatric age in Maldivians, phenomena already observed in other developing countries of the same geographical area [19,20].

## 2. Materials and Methods

### 2.1. Study Design

This study was conducted according to the guidelines in the Declaration of Helsinki, and the institutional review board of the University of Milano–Bicocca (n 429, 28 January 2019) approved all procedures involving human subjects. Before inclusion, the study and its purpose were described to the participants’ parents, who were invited to ask questions about the study and to provide their informed consent. The mayor and the principal as well provided their consent to participate in the study.

A team of scientists and nutritionists was formed for this study; the field team camped at the UNIMIB Marine Research and High Education Center (MaRHE) in October 2018 and performed the investigation at Maghoodoo Public School.

The University of Milano–Bicocca has an outpost in the Maldives in the Faafu Atoll, Maghoodoo Island. This island is not touristy; the inhabitants (mainly fishers) live and work according to local customs. There is only one school and is attended by all the island’s students. The population consumes local and traditional foods with the addition of supplies that arrive from the Asian continent, 1–2 times per week by ferry boat.

For the present study a total of 145 students were recruited consecutively from 22 October to 30 October 2018.

### 2.2. Data Collection

#### 2.2.1. Assessment of Health Status

All the participants of the study underwent anthropometric measurements. Height (Seca 206, Hamburg, Germany), weight (Seca 703), and waist circumference were measured in light clothing. Weight was approximated to the nearest 100 g and height to the nearest 1 cm. Body mass index (BMI) was calculated as weight (kg)/height (m)^2^. PERCBMI and BMI *z*-scores were calculated using the Centers for Disease Control and Prevention charts available at http://www.cdc.gov/nchs/ (accessed on 13 May 2002). Weight class was defined according to the International Obesity Task Force classification, distinguishing among normal weight (NW) and overweight (OW, BMI percentile > 85°). Waist circumference was measured to the nearest 1 cm by a nonelastic flexible tape in standing position. Waist-to-height ratio (WtHr) was calculated by dividing waist circumference by height. BP was measured using an automated oscillometric device (Omron M3, Omron Healthcare, Kyoto, Japan). Measurements were performed after at least 5 min of rest. BP measures were taken 3 times (at 3–5 min intervals), and systolic BP (SBP) and diastolic BP (DBP) percentiles and *z*-scores were calculated according to the nomograms of the National High Blood Pressure Education Program Working Group on High Blood Pressure [21].

The children and adolescents were classified according to the mean of the 3 measurements, as follows: normotensive (NT) if both SBP and DBP percentiles were <90th (*z*-score 1.282) and hypertensive (HT) if SBP and/or DBP percentiles were ≥90th.

#### 2.2.2. Assessment of Nutritional Habits

Data were collected with a questionnaire developed for this study filled by the parents before the anthropometric measurements. The questionnaire was administered in the local language with the help of interpreters (Appendix A).

The questionnaire was divided into three parts: the first part investigated the preweaning period, i.e., duration of breastfeeding, and/or type of milk used. The second part was related to weaning, asking which foods (and when) were introduced in the diet. The questions were formulated on the basis of guidelines for the first 1000 days and on complementary feeding [22,23]. The third part dealt with children’s eating habits at the time of the investigation (consuming certain foods, particularly sweet and salty foods and sweetened drinks). This part was formulated on the basis of a WHO survey on Maldivian populations regarding risk factors for NCDs [24]. The collection of these data was also important since it known that excessive salt and sugar intake can contribute to the onset of hypertension in children, predominantly in subjects with excess weight [25,26].

Regarding food (fruit, vegetables, fish, meat, pulses, rice, eggs), parents were asked how often, on average, their child ate each food, with three possible answers: 0–2 times/day, 3–4 times/day, 6 or more times/day. The questions on sweet and salty foods and sweetened drinks asked the parents to define the average consumption of each food, with three possible answers: 0–once/day, 2–3 times/day, 4 or more times/day.

### 2.3. Statistical Analysis

Anthropometric and blood pressure measures were considered in their original version (raw data) and transformed into *z*-scores according to the references of the International Obesity Task Force and the CDC growth charts, respectively. The derivation of *z*-scores was implemented according to the parameters in the reference tables made available from the WHO and the consensus [27].

Descriptive statistics on categorical variables were obtained by relative frequencies, and comparison across groups was performed by chi-square test. Descriptive statistics on continuous variables (including *z*-scores) were analyzed by mean, standard deviation, and percentiles, and comparison across groups was assessed by *t*-test.

The relationship among blood pressure percentiles and factors that may have an influence on blood pressure data was investigated by a linear regression model.

Data management and analysis were carried out by using the software STATA version 16 (StataCorp, TX, USA).

## 3. Results

Table 1 shows the anthropometric and clinical characteristics of the students at recruitment, subdivided according to gender. The mean age was 9.37 (SD = 2.97), and 50.30% were male. Excess weight was present in 12.50% and 15.07% of males and females, respectively. The distribution of PERC BMI versus age is reported in Figure 1; a significant increase in PERC BMI with age was detected in females (*p* < 0.0001, panel b) but not in males (panel a). Similarly, for BMI *z*-score there was an appreciable difference between genders, showing a tendency to augment with age in the female group, with a regression coefficient estimate equal to +0.211 (95% confidence interval 0.117–0.305, *p*-value < 0.0001).

During the collection of the anthropometric data, we also evaluated the blood pressure of the enrolled subjects. Elevated BP was detected in 15.18% of the population, 8.33% and 6.85% of the male and female population, respectively (Table 1). Among the variables, only the SBP *z*-score was significantly different according to the gender (46.66 ± 24.77 in males vs. 36.82 ± 25.34 in females, *p* = 0.022).

The distribution of the maximum value between SBP *z*-score and DBP *z*-score did not change as a function of BMI *z*-score either in males or in females.

To analyze the eating habits of the pediatric population of the study, we asked the parents to fill in a questionnaire, obtaining a response from 134 parents (93.7%). The results are summarized in Table 2 and Table 3.

Table 2 reports the answers to the questions about breastfeeding and the complementary feeding (CF) period. Early breastfeeding is critical for a child’s health and growth. Almost all the mothers (97.01%) breastfed their babies for a period between 12 and 24 months, and about a third (36.16%) even beyond 24 months. However, it is interesting to note that the practice of exclusive breastfeeding decreased significantly as the age of the child increased, since 83.58% admit the use of other types of milk.

All types of foods included in the questionnaire were used during CF by the population, except for butter (28.36%) and cheese (34.33%) because they do not belong to their culinary tradition. No gender differences were detected regarding breastfeeding or CF.

Our data indicate that 48.51% of the evaluated families introduced sugar between 6 and 12 months, and even 21.64% within the first six months of the child’s life.

During CF, salt was introduced between 6 and 12 months by 44.77%, whereas in the first 6 months by 31.35% of families. Even in this case no gender differences were detected.

The third part of the questionnaire aimed to understand children’s eating habits, in particular, analyzing the frequency of certain food consumption (Table 3).

The majority of children and adolescents (70.15%) consumed less than two portions of fruit per day with a significant difference between gender (84.06% and 55.38% males and females, respectively, *p* < 0.0001). Interestingly, 40% of females ate 3–4 servings of fruit per day. In contrast, most students (71.64%), with no gender differences, only consumed less than two servings of vegetables per day. The low fruits and vegetables intake might be due to the unavailability of wild plants and seasonal fruits on the island.

Consumption of other foods (fish, meat, legumes, rice, and eggs) was instead in line with international guidelines [28], without gender differences except for pulses, which was slightly more abundant in males (56.52% and 49.23% males and females, respectively, *p* < 0.001).

For sweet snacks, 30.6% of the students consumed 2–3 servings per day and 8.21% well over 4 per day. Similarly, for sugary drinks (SSB), 32.84% of the students consumed 2–3 servings per day and 11.19% well over 4 per day. In both cases, no gender difference was observed.

There is also a concern regarding the consumption of salty snacks, since 23.13% of students reported consuming 2–3 servings per day and 4.48% far beyond 4 per day.

In the linear regression model (Table 4), the only factor significantly associated with the BP *z*-score was consumption of sweet snacks in the female population (*p* = 0.029). Neither salty snack consumption nor the timing of salt or sugar introduction into the diet was associated with the outcome.

## 4. Discussion

This research aimed to analyze how a population, which had no relations with the Western diet up to 10 years ago, began to change eating habits, increasing the intake of processed foods, with a consequent surge in noncommunicable diseases (obesity, hypertension, etc.).

The island of Magoodoo can be regarded as a good example of these changes since, in the last 10 years, sweet and salty snacks, sugary drinks, and energy drinks have been imported and sold in the two little shops present on the island, increasing the exposure to junk foods, mainly in the pediatric population.

Indeed, it is well known that the processes and ingredients used in the production of energy-rich, highly processed, and packaged foods are designed to create products that are highly profitable (low-cost ingredients, long shelf life, emphatic branding), convenient (ready-to-eat), and hyperappetitive. These foods, generally characterized by high calories, added sugars, sodium, and unhealthy fats, and also low in fiber, protein, and micronutrients, have extensive adverse effects on human health and the environment (due to associated carbon emissions and water consumption) [29].

In our study, excess weight was mostly present in females. These results are slightly lower than those observed in the 2014 screening of all Maldivian grade-1 students, in which 22% of the children were overweight and obese. The WHO STEPS survey reported a general trend of increasing overweight and obesity with age in Malé population, specifically 42.30% of women were overweight (BMI ≥ 25) and 14.50% were obese (BMI ≥ 30) [24].

It is now clearly recognized that the Maldives are facing a nutritional transition and a severe public health risk, as demonstrated by the high burden of overweight and obesity [30]. Overweight often predisposes to many diseases reducing the quality of life, such as hypertension, dyslipidemia, and diabetes [31], and childhood obesity can progress into the adult form [32,33,34]. In addition, childhood hypertension is an established predictor of hypertension and organ damage in adults, an underestimated problem in developing countries [2,8,9,15].

Our data showed that 15.18% of the studied population were subjects with elevated BP, only partially associated with the weight class. These data parallel those reported by other studies performed both in developed and developing countries, where pediatric hypertension was approximately 14% [35,36,37,38]. However, our survey data are even more worrisome when referring to the Maldivian adult population. A WHO survey performed in 2011 reported that the prevalence of adult hypertensive subjects (SBP ≥ 140 mmHg and/or DBP ≥ 90 mmHg) was 16.6% (19% among men and 14.3% among women) [24].

A rapidly growing number of epidemiological and clinical studies have recently provided evidence that very early exposure to various external factors, and in particular the diet, during prenatal and early childhood development has decisive and durable influences on the subsequent development of obesity, high blood pressure, and associated cardiovascular disease [30]. Particular attention should also be given to the first 1000 days of life from conception, in order to prevent unfavorable epigenetic changes. For this reason, we thought it was appropriate to administer a questionnaire to the parents of our children and adolescents sample in order to analyze their eating habits since birth: breastfeeding, the introduction of complementary foods, and the current diet. Early breastfeeding is critical for a child’s health and growth, and almost all mothers breastfeed their babies for a period between 12 and 24 months.

The introduction of CF is recommended in the period characterized by rapid growth and development, when children are susceptible to nutrient deficiencies and excesses [22]. All types of foods included in the questionnaire were used during CF by the population, (except for butter and cheese). The promotion of exclusive breastfeeding during the first months of life, and the subsequent correct introduction of CF, was made possible by the nutrition education provided to the mothers through the child-friendly hospital initiative and the awareness created in the territory [2].

Our questionnaire gave greater importance to the timing of salt and sugar introduction in the child’s diet. It is known that the prevalence of essential arterial hypertension in children and adolescents has grown considerably in the last few decades, and eating habits is one of the causes of this cardiovascular risk factor [25,26].

Similar data were revealed by Dallazzen et al. [39] in Brazilian municipalities with low socioeconomic status. It is known that children have an innate preference for salty and sweet foods, and early exposure to these foods can also further increase their preference into adulthood [40]. These products are nutritionally useless and can undermine the consumption of staple foods in a healthy, balanced diet [39].

In our sample, the introduction of sugar and salt occurred too early compared to the timing recommended by experts. Previous studies stated that salt and sugar should not be added to CF; intake of free sugars (sugars added to foods and beverages by the manufacturer, cook, or consumer, and sugars naturally present in syrups and juices) should be minimized, and SSB avoided [22,23]. Children who consume SSB have poorer diet quality and higher total energy intake than children who do not consume SSB. Interventions for obesity and chronic disease should replace SSB with water and improve other aspects of diet quality related to SSB consumption [41]. This intervention becomes even more important since it is now known that excessive salt and sugar intake can contribute to the hypertension onset in children, particularly in subjects with excess weight [25].

The STEPS survey, reported that, among men, 43.80% ate one to two servings of fruits and/or vegetables compared to 44.90% of women. An inadequate diet in fruits and vegetables is an independent risk factor for cardiovascular disease and cancer, whereas it has been demonstrated by epidemiological studies that increasing fruit and vegetable quantity in the diet can reduce the risk of the abovementioned pathologies [42]. In addition, increasing the amount of fruits and vegetables in the diet is likely to reduce fat consumption, thereby reducing the risk of diabetes, cardiovascular disease, stroke, and hypertension. For these reasons, steps should be taken to facilitate the availability of locally grown fruits and vegetables in the regional markets. A policy change aimed to reduce prices could also be an intervention to increase the use of fruits and vegetables in the regular diet of ordinary people [24].

Consumption of other foods (fish, meat, legumes, rice, and eggs) was instead in line with international guidelines [28] without sex differences except for pulses.

Thirty percent of the students consumed more than 2–3 servings per day of SSB. This is an alarming finding given that the intake of free sugars, particularly in the form of SSB, increases the overall energy intake and may reduce the intake of more nutritionally adequate calorie-containing foods, thus leading to unhealthy diet, weight gain, and increased risk of NCDs [9,10,11,15,16]. For these reasons, in April 2015, the WHO published the Guidelines on Sugar Intake for Adults and Children [43]. The guideline recommends reducing free sugars intake to less than 10% of total energy intake, i.e., about 12 teaspoons of sugars per day. In addition, a further reduction below 5% of total energy intake, or about 6 teaspoons per day, is recommended for additional health benefits. Added sugars contribute to an energy-dense but nutrient-poor diet, and increase the risk of developing obesity, cardiovascular disease, and hypertension. Although added sugars can be most likely safely consumed in low amounts as part of a healthy diet with fresh food consumption, few children reached this balance, making this a significant public health goal [44].

In addition, salt consumption was also assessed to be high, as compared to the estimated age-standardized sodium intake (g/d) reported as 3.31 g of sodium in 2010 [45].

In summary, it is clear from our data that the children and adolescents in the study have remarkably changed their eating habits compared to their parents’ ones, shifting increasingly more toward a diet similar to the Western diet, with an increase in the consumption of packaged products and a decrease in the intake of fresh or traditional foods. As observed in Western countries, the rise of “diseases of affluence”, including obesity, diabetes, and cardiovascular disease, can be regarded as a consequence of the profound diet and lifestyle changes over the past century [17,46].

We are aware that there are limitations to the present study. The methods of student dietary intake assessment have not been previously validated, and there may be inaccuracies in the maternal reporting. Furthermore, the studied population lives on the same atoll. Still, we believe it is representative of the pediatric-age Maldivian population, since the variations in the diet are mainly due to the exposure to Western imported foods that arrived on the island in the last decade. In addition, it must also be underlined that this is not a tourist atoll, and the presence of the MaRHE Center has not altered at all the food habits of the population living on the island. A minor limitation is the lack of blood sampling, since it would be extremely interesting to evaluate parameters such as cholesterol, triglycerides, and blood sugars.

Nevertheless, to our knowledge, this is the first study done on a genetically homogenous Maldivian children and adolescents population, which not only detects the anthropometric characteristics but also reports the feeding habits from breastfeeding to CF and pediatric age.

## 5. Conclusions

Our data indicate that a Maldivian pediatric population recently approaching Western diet shows excess weight and elevated BP values similar to westerners. Additionally, fruits and vegetables, and sweet and sugary drinks consumption are alarming, suggesting that excess weight and hypertension in this population might be associated with unrestrained consumption of energy-rich, highly processed, packaged foods. A nutrition education approach might help to reduce the cardiovascular risk.

In addition, it is imperative to promote health in schools as an effective treatment, using an approach based on lifestyle modifications. Indeed, this approach has been found to be the most effective in reducing cardiovascular risk worldwide [47]. National policies on significant changes in the food environment, i.e., where people purchase and consume food, are critical to prevent rapid increases in unhealthy food intake and to avoid further increments in obesity and cardiovascular diseases in the population [48].

## Figures and Tables

**Figure 1 ijerph-19-15728-f001:**
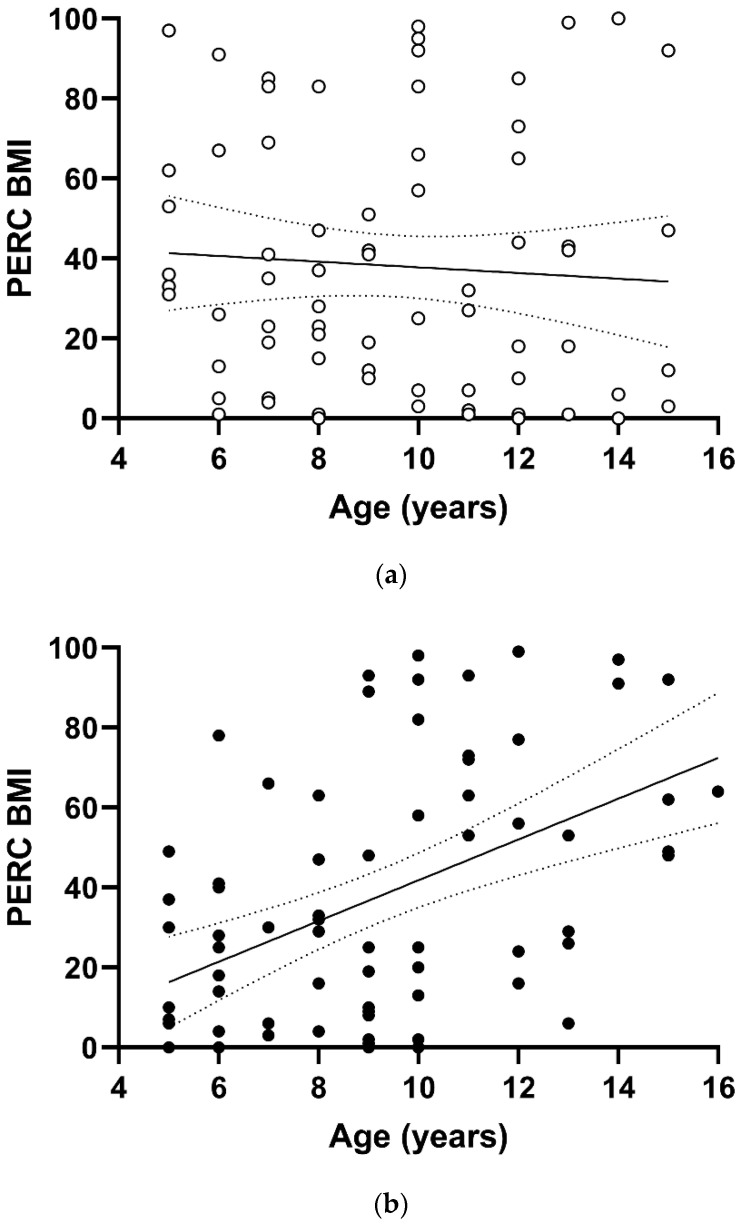
Scatterplot of PERC BMI versus age in groups defined by gender: (**a**) male; (**b**) female.

**Table 1 ijerph-19-15728-t001:** Baseline characteristics of the 145 enrolled subjects subdivided according to gender.

	Total (*n* = 145)														
	Male							Female							
	*n* = 72 (50.3%)							*n* = 73 (49.7%)							
	Mean	SD	Median	p25	p75	Min	Max	Mean	SD	Median	p25	p75	Min	Max	*p*-Value
Age, years	9.53	2.89	9.47	7.21	11.82	4.70	15.42	9.20	3.05	9.05	6.46	10.97	4.58	15.59	0.502
Height, m	1.32	0.17	1.30	1.20	1.43	1.02	1.75	1.28	0.17	1.27	1.12	1.43	0.99	1.63	0.146
Height *z*-score	−0.66	0.96	−0.70	−1.34	0.01	−3.13	1.12	−0.89	1.14	−0.89	−1.54	−0.11	−4.67	1.64	0.182
Weight, kg	30.66	15.18	25.65	20.51	35.55	15.51	105.30	29.50	14.75	24.30	18.00	39.10	12.60	78.40	0.640
BMI, kg/m^2^	16.73	4.18	15.65	14.46	17.54	12.94	40.57	16.81	4.29	15.14	13.96	19.08	12.09	30.39	0.904
BMI *z*-score	−0.48	1.32	−0.48	−1.39	0.39	−2.95	2.69	−0.54	1.36	−0.54	−1.34	0.32	−3.71	2.23	0.814
Weight class															0.895
NW, *n* (%)	63 (87.50)							62 (84.93)							
OW, *n* (%)	9 (12.50)							11 (15.07)							
Waist, cm	58.51	10.26	56.00	51.70	61.50	48.00	111.80	57.60	10.09	54.00	49.00	64.00	43.40	91.50	0.618
WtHr	0.40	0.07	0.43	0.41	0.45	0.00	0.69	0.44	0.05	0.43	0.42	0.45	0.34	0.56	0.233
WtHr > 0.5, *n* (%)	9.72							13.70							0.457
SBP, mm Hg	99.29	10.11	99.00	92.00	104.00	80.00	124.00	94.52	9.79	93.00	89.00	101.0	67.00	121.0	0.005
SBP *z*-score	46.66	24.77	45.92	25.44	65.92	0.92	97.19	36.82	25.34	28.49	17.80	52.18	0.52	92.99	0.022
DBP, mm Hg	63.05	7.05	63.0	58.00	67.50	49.00	70.00	61.94	6.83	62.00	58.00	66.00	45.00	80.00	0.338
DBP *z*-score	61.64	20.16	62.88	45.73	78.49	12.61	97.52	58.70	19.81	58.59	43.49	72.31	16.82	99.32	0.427
Blood pressure category															0.736
NT, *n* (%)	66 (91.67)							68 (93.15)							
HT, *n* (%)	6 (8.33)							5 (6.85)							

BMI = body mass index, NW = normal weight, OW = overweight (BMI > 85 percentile), WtHr = waist-to-height ratio, SBP = systolic blood pressure, DBP = diastolic blood pressure, NT = normotensive, HT = hypertensive.

**Table 2 ijerph-19-15728-t002:** Length of breastfeeding and complementary foods introduction in the whole population and in groups defined by gender.

		Total		Male		Female	
	*n*	(%)	*n*	(%)	*n*	(%)	*p*-Value
	134	100.00	69	51.49	65	48.51	
Breastfeeding	130	97.01	66	95.65	64	98.46	0.339
Breastfeeding time							0.282
0–6] months	17	12.69	9	13.04	8	12.31	
6–12] months	11	8.21	3	4.35	8	12.31	
12–24] months	59	44.03	33	47.83	26	40.00	
>24 months	47	35.07	24	34.78	23	35.38	
Other milk	54	83.58	58	84.06	54	83.08	
Complementary foods (type)							
Flour and cereals	107	79.85	53	76.81	54	83.08	0.366
Fruits and vegetables	131	97.76	67	97.1	64	98.46	0.595
Adult food	114	85.07	57	82.61	57	87.69	0.409
Meat	109	81.34	56	81.16	53	81.54	0.757
Fish	127	94.78	64	92.75	63	96.92	0.436
Butter	38	28.36	15	21.74	23	35.38	0.089
Cheese	46	34.33	20	28.99	26	40.00	0.199
Eggs	117	87.31	58	84.06	59	90.77	0.332
Sugar introduction (age)	132	100.00	68	51.52	64	48.48	0.149
0–6] months	29	21.97	20	29.41	9	14.06	
6–12] months	65	49.24	32	47.06	33	51.56	
12–24] months	26	19.70	13	19.12	13	20.31	
>24 months	12	9.09	3	4.41	9	14.06	
Salt introduction (age)	120	100.00	62	51.67	58	48.33	0.099
0–6] months	42	35.00	24	38.71	18	31.03	
6–12] months	60	50.00	30	48.39	30	51.72	
12–24] months	15	12.50	8	12.90	7	12.07	
>24 months	3	2.50	0	0.00	3	5.17	

0–x] = children of x months are included.

**Table 3 ijerph-19-15728-t003:** Analysis of the children’s eating habits at the time of the visit in the whole population and in groups defined by gender.

Dietary Habit	Total	Male	Female	
	*n* = 134	*n* = 69	*n* = 65	
	(%)	(%)	(%)	*p*-Value
Fruit (times/day)				<0.0001
0–2	70.15	84.06	55.38	
3–4	24.63	10.14	40.00	
>5	2.24	2.90	1.54	
NA	2.99	2.90	3.08	
Vegetables (times/day)				0.106
0–2	71.64	76.81	66.15	
3–4	18.66	11.59	26.15	
>5	3.73	2.90	4.62	
NA	5.97	8.70	3.08	
Fish (times/week)				0.132
0–2	33.58	34.78	32.31	
3–4	35.82	27.54	44.62	
>5	26.87	31.88	21.54	
NA	3.73	5.80	1.54	
Meat (times/week)				0.885
0–2	67.91	68.12	67.69	
3–4	19.40	17.39	21.54	
>5	4.48	4.35	4.62	
NA	8.21	10.14	6.15	
Pulses (times/week)				0.009
0–2	52.99	56.52	49.23	
3–4	26.12	14.49	38.46	
>5	11.94	15.94	7.69	
NA	8.96	13.04	4.62	
Rice (times/week)				0.325
0–2	10.45	7.25	13.85	
3–4	15.67	14.49	16.92	
>5	66.42	69.57	63.08	
NA	7.46	8.70	6.15	
Egg (times/week)				0.119
0–2	64.18	71.01	56.92	
3–4	26.12	20.29	32.31	
>5	0.00	8.70	0.00	
NA	9.70	0.00	10.77	
Sweet snack (times/day)				0.669
0–1	52.99	55.07	50.77	
2–3	30.60	26.09	35.38	
>4	8.21	8.70	7.69	
NA	8.21	10.14	6.15	
SSB (times/day)				0.904
0–1	47.01	46.38	47.69	
2–3	32.84	30.43	35.38	
>4	11.19	11.59	10.77	
NA	8.96	11.59	6.15	
Salty snack (times/day)				0.201
0–1	61.94	56.52	67.69	
2–3	23.13	21.74	24.62	
>4	4.48	7.25	1.54	
NA	10.45	14.49	6.15	

NA = no reply; SSB = sugary beverages.

**Table 4 ijerph-19-15728-t004:** Multivariable linear regression models on the relationship between blood pressure (*z*-score) and possibly correlated variables in groups defined by gender.

Male					Female				
Variable BP, *z*-Score	Estimate	95% CI		*p*-Value	Variable BP, *z*-Score	Estimate	95% CI		*p*-Value
Age, years	−0.02	−0.06	0.02	0.286	Age, Years	−0.02	−0.06	0.02	0.291
BMI, *z*-score	−0.04	−0.13	0.05	0.333	BMI, *z*-score	0.03	−0.05	0.11	0.490
Salt introduction	0.12	−0.23	0.47	0.504	Salt Introduction	0.13	−0.16	0.41	0.363
Sugar introduction	−0.02	−0.39	0.35	0.918	Sugar Introduction	−0.09	−0.42	0.24	0.574
Adult food	−0.03	−0.30	0.25	0.847	Adult food	−0.22	−0.54	0.10	0.162
Salty snacks	−0.05	−0.24	0.14	0.614	Salty Snacks	−0.12	−0.30	0.06	0.191
Sweet snacks	−0.01	−0.20	0.17	0.886	Sweet Snacks	0.25	0.03	0.48	0.029

BMI, body mass index; CI, confidence interval; BP, blood pressure.

## Data Availability

The datasets used and/or analyzed during the current study are available from the corresponding author on reasonable request.

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
