# Peer review of "Health Status and Nutritional Habits in Maldives Pediatric Population: A Cross-Sectional Study"

_ijerph, 2022, doi:10.3390/ijerph192315728_

Round 1

Reviewer 1 Report (Previous Reviewer 2)

Thank you to the authors for the effort in addressing the previous comments. There are still issues to be addressed as follow prior to acceptance:

1) Abstract: There is a lack of description of the methods

2) Discussion: This section can be improved by consolidating the key findings into main paragraphs. Short paragraphs such as line 253-255, line 269-270, line 278 to 282, line 283 to 286, line 296 to 300, line 301 to 306, line 307 to 310, line 355 to 356) should be integrated into the main paragraphs for succinct discussion.

3) Conclusion: Move the limitation and strengths of study to the last section of Discussion section

4) Format: Do check and ensure all decimal points for data reporting throughout the manuscript

5) It is highly encouraged to use English editing service to proof-read the manuscript

Author Response

UNIVERSITY OF milano-bicocca

SCHOOL OF MEDICINE AND SURGERY

Via Cadore, 48  -  20900   Monza (MB) - ITALY

[email protected]

Monza, 11 November 2022

Dear Editor,

Please find enclosed the revised version of the manuscript entitled “Health status and nutritional habits in Maldives pediatric population: a cross-sectional study by Emanuela Cazzaniga, Antonina Orlando, Annalisa Terenzio, Carlotta Suardi, Chiara Mognetti, Francesca Gennaro, Laura Antolini and Paola Palestini.

We modified the article according to the valuable observations of the two reviewers.

Point by point answers are reported below.

Reviewer 1

Thank you to the authors for the effort in addressing the previous comments. There are still issues to be addressed as follow prior to acceptance:

1) Abstract: There is a lack of description of the methods

As suggested, we have added some details.

2) Discussion: This section can be improved by consolidating the key findings into main paragraphs. Short paragraphs (such as line 253-255, line 269-270, line 278 to 282, line 283 to 286, line 296 to 300, line 301 to 306, line 307 to 310, line 355 to 356) should be integrated into the main paragraphs for succinct discussion.

As suggested, we have revised the discussion section.

3) Conclusion: Move the limitation and strengths of study to the last section of Discussion section

Thank you for the suggestion, limitation and strengths have been moved in Discussion section

4) Format: Do check and ensure all decimal points for data reporting throughout the manuscript

We had revised all data reported.

5) It is highly encouraged to use English editing service to proof-read the manuscript

We provide an English revision of the manuscript.

We thank the editorial board and the Reviewers for their constructive comments that have improved the quality of this manuscript and look forward to acceptance.

Sincerely yours

Prof. Paola Palestini

School of Medicine and Surgery

University of Milano-Bicocca (Monza, ITALY)

Reviewer 2 Report (Previous Reviewer 3)

No further comments as the manuscript was improved again.

Author Response

UNIVERSITY OF milano-bicocca

SCHOOL OF MEDICINE AND SURGERY

Via Cadore, 48  -  20900   Monza (MB) - ITALY

[email protected]

Monza, 11 November 2022

Dear Editor,

Please find enclosed the revised version of the manuscript entitled “Health status and nutritional habits in Maldives pediatric population: a cross-sectional study by Emanuela Cazzaniga, Antonina Orlando, Annalisa Terenzio, Carlotta Suardi, Chiara Mognetti, Francesca Gennaro, Laura Antolini and Paola Palestini.

We modified the article according to the valuable observations of the two reviewers.

We thank the editorial board and the Reviewers for their constructive comments that have improved the quality of this manuscript and look forward to acceptance.

Sincerely yours

Prof. Paola Palestini

School of Medicine and Surgery

University of Milano-Bicocca (Monza, ITALY)

This manuscript is a resubmission of an earlier submission. The following is a list of the peer review reports and author responses from that submission.

Round 1

Reviewer 1 Report

Overall it is a good study may be a giving new data to Maldives to see the changes in the social structure.  

Objective:

1.       Line 77 - Very broad it should be explained in the method section in details. What health and nutrition indicators were studied and their definitions? This will improve the clarity and the focus of the manuscript.

Methods:

1.       Include the sample size calculation, which is missing.

2.       Data collection period – Need to be added, which will provide more validity

3.       Study population, study site and sampling are also missing in the method section.

4.       Line 113 – need to explain the BMI cut-off for overweight etc. Usually WHO references is used for better comparisons. However, that is up to the author. CDC and WHO, both are valid. 

5.       Better to add the quality measures adopted for measurements. What protocols are adopted etc.  What is the specific reason to take measurement while they were fasting?

Results

6.        Line 123 – Is it a cohort or cross-sectional data?

7.       Line 124 – What do you mean by excess weight? Is it overweight or obese or both?

8.       Need to add basic characteristics table at the beginning to identify the sociodemographic characteristics of the sample.

9.       Table 1 – media should be corrected to median

10.   Line 150 – It is better to provide prevalence data on overweight, obesity etc. then comment and discuss with the available data in the country. Discussion points are nonspecific, better to be more precise.

11.   Line 171 – Could not locate prevalence data to support this statement. I feel it is better to provide prevalence data on high BP with cut-off, high waist with defined cut-off levels. Then it is easy to compare with the other population groups.

12.   Regression data – need to include other variables into the model; BMI, Waist etc. If not valid for the model, please explain in the data analysis section.

Conclusion

13.   Better to be focused and confined to the results of the study. Some are very generic and better to shift to discussion.  

Reference

Need to re look at and correct according to the Journal style. Need the Author of this publication. There are many like this.

E.g.  . Nutritional Status of mothers and young children in Maldives -How can it be improved? 1999 http://www.nutrisurvey.de/sur- 369 veys/maldives.pdf

Author Response

Reply to Reviewer #1

Objective:

  1. Line 77 - Very broad it should be explained in the method section in details. What health and nutrition indicators were studied and their definitions? This will improve the clarity and the focus of the manuscript.

As suggested by reviewer 1, we have revised the introduction section.

Methods:

  1. Include the sample size calculation, which is missing.
  2. Data collection period – Need to be added, which will provide more validity
  3. Study population, study site and sampling are also missing in the method section.

Children were enrolled consecutively. A specification was added in the text (2.1 Study Design section).

  1. Line 113 – need to explain the BMI cut-off for overweight etc. Usually WHO references is used for better comparisons. However, that is up to the author. CDC and WHO, both are valid.
  2. Better to add the quality measures adopted for measurements. What protocols are adopted etc. What is the specific reason to take measurement while they were fasting?

We have inserted the explanation of these points in 2.2.1 paragraph.

Results

  1. Line 123 – Is it a cohort or cross-sectional data?

As suggest, we change “cohort” with “group”.

  1. Line 124 – What do you mean by excess weight? Is it overweight or obese or both?

We decided, in our analyzes, to combine overweight children (BMI percentile> 85th) and obese children (BMI percentile> 95th) .There is a lot of data from the literature confirming that there is a connection between being overweight or obese and cardiovascular disease. Children who were not only obese but also overweight or in the upper quartiles for the variable adiposity had higher blood pressure levels and a lipid profile indicative of an increased risk of developing atherosclerosis. (E Jokinen Obesity and cardiovascular disease Minerva Pediat. 2014, 67(1):25-32.; Additional cardiovascular risk factors associated with excess weight in children and adolescents: the Belo Horizonte heart study. Ribeiro RQ, et al. Arq Bras Cardiol. 2006. Arq Bras Cardiol. 2006 Jun;86(6):408-18.)

  1. Need to add basic characteristics table at the beginning to identify the sociodemographic characteristics of the sample.

The only sociodemographic characteristics of the sample that we have are included at the beginning of table 1. All the students are born and live on Magoodhoo Island.

  1. Table 1 – media should be corrected to median

Sorry for the mistake Table 1 has been correct

  1. Line 150 – It is better to provide prevalence data on overweight, obesity etc. then comment and discuss with the available data in the country. Discussion points are nonspecific, better to be more precise.

We have revised the discussion section, we hope that it will be more clarifying.

  1. Line 171 – Could not locate prevalence data to support this statement. I feel it is better to provide prevalence data on high BP with cut-off, high waist with defined cut-off levels. Then it is easy to compare with the other population groups.

Unfortunately, these are the only Maldivian data published.

  1. Regression data – need to include other variables into the model; BMI, Waist etc. If not valid for the model, please explain in the data analysis section.

BMI was included in the models though the standardized measure BMI z-score (according to an independent reference standard) as measure of excess weight. The use of the raw BMI is not suitable in a growing population (similarly to the raw measures of blood pressure) to assess excess weight. WtHr was not included in the model (but considered only for a descriptive purpose) since we are still in the absence of a reference standard in a growing population to measure excess of WtHr, although WtHR (similarly to BMI) is a relative measure (with respect to height).

Conclusion

  1. Better to be focused and confined to the results of the study. Some are very generic and better to shift to discussion.  

As suggest, we divided results and discussion and rewrite the conclusion.

Reference

Need to re look at and correct according to the Journal style. Need the Author of this publication. There are many like this.

E.g.  . Nutritional Status of mothers and young children in Maldives -How can it be improved? 1999 http://www.nutrisurvey.de/sur- 369 veys/maldives.pdf

As the reviewer will have noticed, some references are not scientific papers but epidemiological data from international sites such as WHO. Therefore, it is not possible to indicate the author.

We thank the editorial board and the Reviewers for their constructive comments that have improved the quality of this manuscript and look forward to acceptance.

Sincerely yours

Prof. Palola Palestini

School of Medicine and Surgery

University of Milano-Bicocca (Monza, ITALY)

Reviewer 2 Report

Thank you for the opportunity to review this article entitled "Health status and nutritional habits in Maldives pediatric population”. There are some issues to be addressed in this paper prior to acceptance for publication as listed in the attached document.

Author Response

Reply to Reviewer #2

Thank you for the opportunity to review this article entitled "Health status and nutritional habits in Maldives pediatric population”. There are some issues to be addressed in this paper prior to acceptance for publication as follows:

1) Abstract

  • There is a lack of description of the methods used and how analyses are performed

between the study variables

  • Line 27-28: Suggest to revise this sentence as the study design in this manuscript is cross-sectional

As request, we have revised the abstract section.

  • Do standardise all decimal points for the reporting of data (percentages, absolute number

etc)

As suggest, we standardize all decimal points.

2) Introduction

  • This section requires more work to present the background of the study and to be

written in a concise and logical manner as follow:

- Start the section with the global relevance to the issues and focus of this

manuscript on health status and nutritional habits

- Combine Paragraph 2 (Line 45 to 51) and Paragraph 4 (Line 59 to 64)

-Combine Paragraph 3 (Line 52 to 58) and Paragraph 6 (Line 74 to 76)

- Line 52 to 54: Include relevant reference to support the sentence and move to the

conclusion section

- Be consistent on the usage of term ‘children’ and ‘pediatric’ as the age range

covered by both terms are different

We thank the reviewer for raising these points that give to us the opportunity to revise the introduction section.

3) Materials and Methods

  • Some specific suggestions for this section as follow to provide details on the conduct of

the study:

-Include the following subsections at the beginning of this subsection: Study Design,

sampling technique (include rationale why the ‘Maghoodoo Public School’ is selected), sample size calculation, study participants’ inclusion and exclusion criteria, ethical consideration

 2.1 Measures: Suggest to rename as ‘data collection’, and introduce subsections for each study variable included in this study, for example: 2.2.1 Assessment of Health Status, 2.2.2 Assessment of Nutritional Habits

- Line 113 to Line 115: Suggest to move this under 2.2.1 Assessment of Health Status. For blood measures, it would be more accurate to write as blood pressure

We thank the reviewer for raising these points that give to us the opportunity to revise the Materials and Methods section.

- For all study variables, state the origin of the equipment/tools, and for the questionnaire, some explanation on the development, validation, piloting and total items included in the questionnaire would be essential. For the questionnaire, under each part, state the number of item asked.

In response to this comment, we have decided to add the questionnaire as supplementary data.

  • Line 120: Explain what is ‘possible explanatory variables’

The sentence “possible explanatory variables” was corrected as “Factors which may influence”.

4) Results and discussion

  • Overall, this section requires more work to present the results clearly hence the suggestion is to split results and discussion into two separate sections in this manuscript. The results and discussion should be reorganised by following the study objectives and focusing on the key findings

As suggest, we split results and discussion in two separated section

  • Some specific comment:

 Line 208 to 209: Suggest to move this sentence to section ‘Materials and Methods’, and explain the rationale on such importance

As suggest we move the sentence in Materials and Methods and explain the rational adding 2 new references.

 5) Conclusion

  • The conclusion section can be shortened by stating the key findings and key recommendations for future (research and practice in Maldives), and include the limitations and strengths of this study

We thank the reviewer for raising these points that give to us the opportunity to revise the Conclusion section

We thank the editorial board and the Reviewers for their constructive comments that have improved the quality of this manuscript and look forward to acceptance.

Sincerely yours

Prof. Palola Palestini

School of Medicine and Surgery

University of Milano-Bicocca (Monza, ITALY)

Reviewer 3 Report

The manuscript reports findings from a cross-sectional study investigating some aspects of children's health status and eating habits in the Maldives.

The manuscript isn't very structured and it's following a standard article format. Moreover, several explanations are missing and also several critical mistakes.

Specific comments:

Title: This manuscript reports findings from a cross-sectional study. Please mention this fact in the title.

lines 77-78: This isn't a very clear aim! Please make it more specific so that it fits with the statistical analyses and resulting findings reported later on.

lines 98-110: Were any of the 3 questionnaires (used in the 3 parts) validated previously? Or were they "developed for this study? Please provide details.

lines 112-121: Which statistical software program was used. Please provide details. Also, isn't it standard to report the significance level used?

lines 114-115: It would be helpful to provide details on how z scores were calculated. Which software?

line 119: You mean "two-sample t-test". Please be precise.

lines 120-121: Percentiles lead to categories. How can linear regression be a suitable model?

lines 122-299: It doesn't work very well to mixed results and discussion, which is often an introduction. Also, there seems not to be much reflection about strength and limitation of results that seems to be based on a convenience sample (as it's not at all clear why a specific school in a specific place on a specific island is of relevance).

line 129: You mean p<0.0001?!

lines 130-131: Why do you compare female and male? You didn't specifiy such an aim.

Table 1: What does "Media" mean? Mean? Confusing table.

Less would be more. Why don't you adhere to the standard reporting format where either mean and SD or median and IQR are reported?

Footnote: "BMI>85" is a disturbing mistake. Why don't you use the same cutoffs as later on in the Results section: 25 and 30?

Figure 1 isn't edited at all but directly taken from the statistical software program used. Please provide plots in a much better quality. Same comment for Figure 2.

Moreover, what is the purpose of showing such messy plots?

Also, please report in the text the estimated slope and the corresponding standard error or confidence intervals and p-value instead of simply copying output directly from the statistical software. Same comment for Figure 2.

lines 148-156: I would prefer that you separated introduction, results, and discussion into 3 sections instead of mixing it up.

Figure 2 shows no significant trends. So why show the 2 plots? A sentence would sentence.

lines 175-187: At best this part belongs to the introduction. It's not results and also not discussion, but partly introducing the study, partly trivial knowledge.

Table 2: It seems the table contains results for 134 participants, 145! Please at least explain why there is a different.

No footnote for Table 2, explaining methods used shortly? Also, what does "0-6]" mean?

lines 195-207: Please make it clear in the text that no significant differences were found when comparing females and males.

Table 3: Why is there a gender difference for pulses? Do you discuss this significant finding at all?

Table 4: Please reformat the table: comma shouldn't be used as decimal separator in English.

More importantly, Table 4 is full of null findings, no associations except for one case that is most likely a chance finding! It would be helpful to mention this fact explicitly in the text.

lines 300-327: This isn't a conclusion but a kind of discussion most of the time. Please reduce the text to a single short paragraph that honestly summarize the limited findings of this study.

Author Response

Reply to Reviewer #3

The manuscript reports findings from a cross-sectional study investigating some aspects of children's health status and eating habits in the Maldives.

The manuscript isn't very structured and it's following a standard article format. Moreover, several explanations are missing and also several critical mistakes. The manuscript isn't very structured and it's following a standard article format. Moreover, several explanations are missing and also several critical mistakes.

Specific comments:

Title: This manuscript reports findings from a cross-sectional study. Please mention this fact in the title.

As suggest we modify the title as proposed

lines 77-78: This isn't a very clear aim! Please make it more specific so that it fits with the statistical analyses and resulting findings reported later on.

As suggest also by reviewer 1, we have revised the introduction section.

lines 98-110: Were any of the 3 questionnaires (used in the 3 parts) validated previously? Or were they "developed for this study? Please provide details.

As suggest, we have decided to add the questionnaire as supplementary data.

lines 112-121: Which statistical software program was used. Please provide details. Also, isn't it standard to report the significance level used?

lines 114-115: It would be helpful to provide details on how z scores were calculated. Which software?

The software was mentioned in the revised version and details on the calculation of z-scores were added.

 line 119: You mean "two-sample t-test". Please be precise.

We wrote regression model since the regression model was used. T test was mentioned in the previous line (2.3 section).

lines 120-121: Percentiles lead to categories. How can linear regression be a suitable model?

We calculated the individual percentile corresponding to each the single raw data, so they are not categories, but continuous data that can be used in a regression model.

This approach was used in:

Genovesi S, Orlando A, Rebora P, Giussani M, Antolini L, Nava E, Parati G, Valsecchi MG. Effects of Lifestyle Modifications on Elevated Blood Pressure and Excess Weight in a Population of Italian Children and Adolescents. Am J Hypertens. 2018 Sep 11;31(10):1147-1155. doi: 10.1093/ajh/hpy096. PMID: 29982339.

Antolini L, Giussani M, Orlando A, Nava E, Valsecchi MG, Parati G, Genovesi S. Nomograms to identify elevated blood pressure values and left ventricular hypertrophy in a paediatric population: American Academy of Pediatrics Clinical Practice vs. Fourth Report/European Society of Hypertension Guidelines. J Hypertens. 2019 Jun;37(6):1213-1222. doi: 10.1097/HJH.0000000000002069. PMID: 31022109.

among others.

lines 122-299: It doesn't work very well to mixed results and discussion, which is often an introduction.

As suggest, we split results and discussion in two separated section

Also, there seems not to be much reflection about strength and limitation of results that seems to be based on a convenience sample

We thank the reviewer for raising these points that give to us the opportunity to revise the Conclusion section

(as it's not at all clear why a specific school in a specific place on a specific island is of relevance).

As suggest, we added the motivation in 2.1 Study design

line 129: You mean p<0.0001?! 

YES it is true!

lines 130-131: Why do you compare female and male? You didn't specifiy such an aim.

As suggest, we add a sentence  (at the end of the Introduction)

Table 1: What does "Media" mean? Mean? Confusing table.

Sorry for the mistake, Table 1 has been correct

Less would be more. Why don't you adhere to the standard reporting format where either mean and SD or median and IQR are reported?

We prefer a more extensive description to enlighten the features of the distribution (symmetry, outliers…) since we didn’t include histograms/boxplots.

Footnote: "BMI>85" is a disturbing mistake. Why don't you use the same cutoffs as later on in the Results section: 25 and 30?

As suggest, we add a sentence (2.2.1 section)

Figure 1 isn't edited at all but directly taken from the statistical software program used. Please provide plots in a much better quality. Same comment for Figure 2.

Moreover, what is the purpose of showing such messy plots?

Also, please report in the text the estimated slope and the corresponding standard error or confidence intervals and p-value instead of simply copying output directly from the statistical software. Same comment for Figure 2.

Figure 2 shows no significant trends. So why show the 2 plots? A sentence would sentence.

In the revised version

  • Figure 1 has novel labels and increased the number of ticks.
  • Figure 2 was removed
  • Model results are added to the text and not embedded in the figures.

Scatter plots are by definition “messy” (or rather on real data), we agree. They are however, are the only one way to explore bivariate data. Tendency lines were in fact added to improve the interpretation.

 lines 148-156: I would prefer that you separated introduction, results, and discussion into 3 sections instead of mixing it up.

As suggest, we had separate Results and Discussion section

lines 175-187: At best this part belongs to the introduction. It's not results and also not discussion, but partly introducing the study, partly trivial knowledge.

We had rewrite the discussion section.

Table 2: It seems the table contains results for 134 participants, 145! Please at least explain why there is a different.

We thank the reviewer for raising this point, sorry it is a mistake, we correct the table and the text.

No footnote for Table 2, explaining methods used shortly? Also, what does "0-6]" mean?

As suggest, we add the footnote for Table 2 

lines 195-207: Please make it clear in the text that no significant differences were found when comparing females and males.

As suggest, we correct the sentences.

Table 3: Why is there a gender difference for pulses? Do you discuss this significant finding at all?

It is difficult to explain this point, maybe this gender difference could be related to a different awareness of food consumption.

Table 4: Please reformat the table: comma shouldn't be used as decimal separator in English.

Sorry for the mistake as suggest, we correct the table 4

More importantly, Table 4 is full of null findings, no associations except for one case that is most likely a chance finding! It would be helpful to mention this fact explicitly in the text.

As suggest, we correct the text.

lines 300-327: This isn't a conclusion but a kind of discussion most of the time. Please reduce the text to a single short paragraph that honestly summarize the limited findings of this study.

We thank the reviewer for raising these points that give to us the opportunity to revise the Conclusion section

We thank the editorial board and the Reviewers for their constructive comments that have improved the quality of this manuscript and look forward to acceptance.

Sincerely yours

Prof. Palola Palestini

School of Medicine and Surgery

University of Milano-Bicocca (Monza, ITALY)

Round 2

Reviewer 1 Report

Most of the comments are done in this version.  Line 110-117, duplicating better to remove. 

Author Response

Dear reviewer,

Sorry, we don't understand the “duplicating”, all methods are reported singularly.

Reviewer 3 Report

Thanks for revising the manuscript, leading to a somewhat improved manuscript. No further comments except that you should check lines 200 and 203 (no full stop and a comma too much) and Figure 1 doesn't look good (please don't simply use poor figures from Stata for publication). Finally, the conclusion is still far too long.

Author Response

Dear Reviewer,

Point by point answers  are reported below:

Thanks for revising the manuscript, leading to a somewhat improved manuscript. No further comments except that you should check lines 200 and 203 (no full stop and a comma too much)

We corrected the sentence.

Figure 1 doesn't look good (please don't simply use poor figures from Stata for publication).

We cannot modify the figure further, could you be more specific in the requirements you want?

Finally, the conclusion is still far too long.

As suggested, we have reduced the conclusion section.